# Associations between relative grip strength and type 2 diabetes mellitus: The Yangpyeong cohort of the Korean genome and epidemiology study

Geon Hui Kim[1], Bong Kil Song [1,2]*, Jung Woon Kim[1¤], Elizabeth C. Lefferts[2], Angelique G. Brellenthin[2], Duck-chul Lee[2], Yu-Mi Kim[3,4], Mi Kyung Kim[3,4], Bo Youl Choi[3,4], Yeon Soo Kim[1,5]

1 Department of Physical Education, College of Education, Seoul National University, Seoul, Republic of Korea, 2 Department of Kinesiology, College of Human Sciences, Iowa State University, Ames, Iowa, United States of America, 3 Department of Preventive Medicine, College of Medicine, Hanyang University, Seoul, Republic of Korea, 4 Institute for Health and Society, Hanyang University, Seoul, Republic of Korea, 5 Institute of Sports Science, Seoul National University, Seoul, Republic of Korea

¤ Current address: Department of Aero Fitness, Korea Air Force Academy, Cheong-Ju, Chungchungbuk-do, Republic of Korea
* bksong@iastate.edu

**Data Availability Statement:** The data used in our study were limited to research purposes only and cannot be made publicly available due to privacy

## Abstract

### Objective

To investigate the association between relative grip strength and the prevalence of type 2 diabetes mellitus (T2DM) independently and in combination with body mass index (BMI) in Korean adults.

### Methods

The cross-sectional study includes 2,811 men and women (age 40 to 92 years old) with no history of heart disease, stroke, or cancer. Relative grip strength was measured by a hand-held dynamometer and calculated by dividing absolute grip strength by body weight. Logistic regression analysis was used to calculate the odds ratios (ORs) and 95% confidence intervals (CIs) of T2DM by sex-specific quintiles of relative grip strength. In a joint analysis, participants were classified into 4 groups: "weak (lowest 20% quintile one) and normal weight (BMI <25.0 kg/m$^2$)", "weak and overweight/obese (BMI $\geq$25.0 kg/m$^2$)", "strong (upper 80% four quintiles) and normal weight" or "strong and overweight/obese".

### Results

Among the 2,811 participants, 371 were identified as having T2DM. Compared with the lowest quintile of relative grip strength (weakest), the ORs (95% CIs) of T2DM were 0.73 (0.53–1.02), 0.68 (0.48–0.97), 0.72 (0.50–1.03), and 0.48 (0.32–0.74) in upper quintiles two, three, four, and five, respectively, after adjusting for BMI and other potential confounders. In the joint analysis, compared with the "weak and overweight/obese" reference group, the odds of

policy. Data are owned and distributed by The Korea Centers for Disease Control and Prevention (KCDC). The data are available from the Division of Epidemiology and Health Index, KCDC (http://cdc.go.kr, Tel; 82-43-719-6745).

**Funding:** This work was partly supported by the Research Program funded by the Korean Centers for Disease Control and Prevention (grant numbers: 2004-E71004-00, 2005-E71011-00, 2006-E71009-00, 2007-E71002-00, 2008-E71004-00, 2009-E71006-00, 2010-E71003-00, 2011-E71002-00, 2012-E71007-00, 2013-E71008-00, 2014-E71006-00, 2014- E71006-01, 2016-E71001-00, grant receiver: BY Choi). There was no additional external funding received for this study. The funders had no role in study design, data collection, and analysis, decision to publish, or preparation of the manuscript.

T2DM [ORs (95% CIs)] was lower in the "strong and overweight/obese" group [0.65 (0.46–0.92)] and the "strong and normal weight" group [0.49 (0.35–0.67)], after adjusting for potential confounders.

## Conclusion

In this cross-sectional study, greater relative grip strength was associated with a lower prevalence of T2DM independent of BMI in Korean adults. Additional prospective studies are needed to determine whether a causal association exists between relative grip strength and T2DM prevalence considering BMI.

## Introduction

Type 2 diabetes mellitus (T2DM) is a growing global health concern, which is associated with a greater risk of mortality [1]. Individuals with T2DM present with insulin resistance and have high blood glucose levels, as tissue no longer effectively uptakes glucose [2]. Skeletal muscle is a major organ for glucose uptake [3], and indeed, lower muscle mass is associated with a greater risk of developing T2DM [4].

Grip strength is a simple, common measure of overall muscular strength and is well known as a prognostic indicator of mortality and various chronic diseases [5, 6]. Previous studies have shown that individuals with low grip strength are at a greater risk of developing T2DM [7–9]. However, grip strength is related to body size, with larger individuals being stronger in general [10, 11]. Studies have shown that relative grip strength, which is grip strength divided by body weight or BMI, may be more strongly associated with metabolic syndrome and cardiometabolic risk factors (e.g., blood pressure, triglyceride, fasting glucose) than absolute grip strength [12–14]. Thus, relative grip strength may have a stronger association with T2DM than absolute grip strength.

Obesity is attributed to insulin resistance by permanently increasing free fatty acids in plasma and reducing glucose uptake by muscle [15]. Many previous studies have elucidated the importance of obesity, assessed by body mass index (BMI) $\geq 30$ kg/m$^2$, as a prominent risk factor for T2DM [16, 17]. Thus, the association between grip strength and T2DM may depend on BMI status. Previous studies investigating the relationship between grip strength and T2DM have controlled for BMI [18], however, few have investigated how the association between relative grip strength and T2DM is varied with BMI status.

It is important to examine the combined effects of muscular strength and overweight/obesity since both are important to develop public health recommendations and policies. For example, if people with overweight/obesity can reduce the risk of T2DM by improving muscular strength, this has a large clinical and public health impact since overweight/obesity is prevalent in most mid- and high-income countries [19]. Further, there is still limited data on the relative contributions of muscular strength versus overweight/obesity to T2DM. Therefore, the purpose of this study is to investigate the relationship of relative grip strength with T2DM using data from the Yangpyeong cohort of Korean Genome and Epidemiology Study. We aim to 1) evaluate the relationship between relative grip strength and the prevalence of T2DM independent of BMI and 2) investigate the combined association of relative grip strength and BMI with T2DM.

## Materials and methods

### Study population

Study participants were from the Yangpyeong cohort, one of the Multi-Rural Community cohorts within the Korean Genome and Epidemiology Study, which is being conducted by the Korea Centers for Disease Control and Prevention to identify risk factors for cardiovascular disease in Koreans (KoGES_CAVAS) [20]. A total 3,351 men and women (aged ≥ 40 years) were enrolled from centers located in Yangpyeong County and completed questionnaires, blood analysis, physical measurements, and fitness tests between 2007 to 2016.

Among the 3,351 participants who participated in the examination, 540 participants were excluded for the following reasons: history of cancer, stroke, and/or heart disease (n = 312); not fasting over 8 hours (n = 20); or missing data on covariates (n = 39) or grip strength (n = 169). In total, 2,811 participants were included in this study. The study was approved by the Institutional Review Board of Hanyang University (IRB No: HYUN IRB 2005–15). The study was conducted in accordance with the Declaration of Helsinki. All patients provided written informed consent, and all data were fully anonymized prior to analysis.

### Grip strength assessment

Grip strength was measured using a handheld dynamometer T.K.K.5401 (Takei, Tokyo, Japan) which has demonstrated both good test-retest reliability and criterion-related validity [21]. Participants experiencing hand or wrist pain or who had previous surgery were excluded from the measurement. Participants were instructed to stand upright with their arm fully extended, slightly away from the side of their body, and to squeeze the dynamometer for 3 seconds as hard as they could, alternating two times per hand with 30 seconds rest between measurements. The highest values from each hand were averaged together and used as the absolute grip strength (kg) [22].

Relative grip strength was calculated by dividing absolute grip strength (kg) by body weight (kg) following earlier studies [11, 12, 23, 24]. Participants were then categorized into sex-specific quintiles of relative grip strength.

### Diagnosis of T2DM

T2DM was defined according to the following criteria: (1) self-reported diagnosis of T2DM by a physician; (2) current use of hyperglycemic medication or insulin; and/or (3) fasting blood glucose over 126 mg/dL [2]. Fasting blood glucose and lipid profile were analyzed using an ADVIA1650 Automatic Analyzer (Siemens, New York, USA) [25].

### Covariates

Participants underwent a clinical examination following an ≥8 hour fast. Weight and height were measured using a standard clinical scale and stadiometer, respectively, and BMI was calculated as weight (kg) divided by height squared ($m^2$). Participants were classified as normal weight, overweight, or obese using the World Health Organization cut-points of <25.0 kg/$m^2$, 25.0–29.9 kg/$m^2$, and ≥30.0 kg/$m^2$, respectively [26].

Resting blood pressure was measured using a sphygmomanometer. After resting for five minutes in the seated position, systolic and diastolic blood pressure were measured at least twice at intervals of one minute. If the two systolic or diastolic blood pressure differed by more than 5 mmHg, additional measures were taken until the last two blood pressure were within 5 mmHg, and the average of the last two blood pressure values was used. Hypertension was defined by combining the questionnaire and blood pressure examination results. A participant

was considered to have hypertension if they indicated a previous physician diagnosis, were currently taking anti-hypertensive medication, or their systolic blood pressure was $\geq 130$ mmHg or diastolic blood pressure $\geq 80$ mmHg [27].

Dyslipidemia was classified by combining the questionnaire and clinical examination results. A participant was considered to have dyslipidemia if they indicated a previous physician diagnosis, were currently using anti-lipidemic medications, or if their serum low-density lipoprotein $\geq 130$ mg/dL, total cholesterol $\geq 200$ mg/dL, high-density lipoprotein $\leq 40$ mg/dL, or triglyceride level $\geq 150$ mg/dL [28].

Participants completed an interview-based questionnaire on demographic characteristics (e.g. age, sex, smoking status, current alcohol drinking, living with family, education level, regular exercise participation), medical conditions (e.g. hypertension, dyslipidemia, T2DM), and family history of diabetes. Smoking status was classified into three categories: never smoker, former smoker, or current smoker. Current alcohol drinking was classified into "yes" or "no". Living with family was indicated as "yes" if the participant had lived with family for the last year, or "no" if the participant had lived alone for the last year. Education level was classified as high school graduate (12 years of education) or not. Regular exercise participation was indicated as "yes" or "no" based on the reported answer to the question, "Do you regularly participate in exercise that elicits sweat?". Participants were classified as having a family history of diabetes if the parents, siblings, or children had been diagnosed or died from diabetes.

## Statistical analysis

Participant characteristics at baseline are expressed as means and standard deviations (SD) for continuous variables and percentages for categorical variables. Baseline characteristics between sex-specific relative grip strength quintiles and T2DM status were compared using ANOVA tests for continuous variables and Chi-squared tests for categorical variables. Logistic regression analysis was used to estimate the odds ratios (OR) and 95% confidence intervals (CI) of T2DM according to the relative grip strength quintiles. The lowest quintile was used as the reference group. Logistic regression was also used to determine linear trends of T2DM by quintiles of relative grip strength. In addition, the ORs for the prevalence of T2DM per SD change were also calculated. All analyses were completed adjusting for sex and age (model 1). In Model 2, we further adjusted for smoking status (never, former, current), current alcohol drinking (yes or no), regular exercise (yes or no), living with family (yes or no), $\geq$high school graduate (yes or no), family history of diabetes (yes or no), hypertension (yes or no), and dyslipidemia (yes or no). Model 3 was additionally adjusted for BMI as a continuous variable. Stratified analyses were conducted to assess whether the association of relative grip strength and T2DM differs for sex, age, smoking status, current alcohol drinking, and regular exercise. Associations between relative grip strength and the OR of T2DM revealed consistent trends across all strata with no significant interaction effects observed (p>0.05).

A joint analysis was conducted by dichotomizing relative grip strength ("weak" [lowest 20%] or "strong" [upper 80%]) and BMI ("overweight/obese" [BMI $\geq 25.0$] or "normal weight" [BMI $< 25.0$]) and categorizing participants into 'weak and overweight/obese', 'strong and overweight/obese', 'weak and normal weight', or 'strong and normal weight' groups. The weak and overweight/obese group was used as the reference group. This categorization was based on the recommendation of the Asian Working Group for Sarcopenia (AWGS) [29]. AWGS recommended using the lower 20th percentile of grip strength of the study population as the cutoff value for low muscle strength before if outcome-based data is unavailable. Although AWGS suggested being defined low grip strength as <26 kg for men and <18kg for women, we cannot use this absolute cutoff value because we used relative grip strength. The

method using the cutoff at the 20th percentile was reported in previous studies [30–32]. A sensitivity analysis was conducted to assess the robustness of the original statistical model by altering exposure categories and covariate definitions. 1) Relative grip strength was categorized into quartiles, and then again into tertiles, to explore exposure-outcome relationships (S1 Table). 2) Absolute grip strength was used instead of relative grip strength. 3) BMI was re-categorized using the cut-off points for the Asian population (normal weight: < 23.0 kg/m$^2$, overweight: 23.0–25.0 kg/m$^2$, and obesity ≥ 25.0 kg/m$^2$) [33]. All analyses were conducted using SAS software version 9.4 (SAS Institute, Inc., Cary, NC), and 2-sided $p$ values <0.05 were considered significant.

## Results

Among the 2,811 participants, 371 were identified as having T2DM. Table 1 shows the baseline characteristics of the participants by quintiles of relative grip strength and T2DM status. Compared to other groups, participants in the highest relative grip strength group were more likely to be younger, have lower BMIs, participate in more exercise, live alone, and have less hypertension and dyslipidemia.

**Table 1. Baseline characteristics of the participants by quintiles of relative grip strength.**

| Characteristics | All | Quintiles of relative grip strength | | | | | P value | T2DM[a] | | P value |
|---|---|---|---|---|---|---|---|---|---|---|
| | | Q1 (Weakest) | Q2 | Q3 | Q4 | Q5 (Strongest) | | Cases | Non-cases | |
| N | 2,811 | 563 | 562 | 562 | 562 | 562 | | 371 | 2,440 | |
| Women, n (%) | 1,755 (62.4) | 351 (62.3) | 351 (62.5) | 351 (62.5) | 351 (62.5) | 351 (62.5) | | 204 (55.0) | 1,551 (63.6) | |
| Age (years) | 60.5 (10.4) | 65.4 (9.6) | 61.9 (9.7) | 60.4 (10.1) | 58.7 (9.8) | 56.4 (10.4) | < .001 | 62.5 (9.4) | 60.3 (10.5) | < .001 |
| Weight (kg) | 61.1 (10.0) | 64.1 (10.0) | 63.2 (10.3) | 62.2 (9.6) | 59.9 (9.2) | 55.9 (8.7) | < .001 | 63.4 (10.4) | 60.7 (9.9) | < .001 |
| BMI (kg/m$^2$)[b] | 24.5 (3.2) | 26.1 (3.4) | 25.3 (3.2) | 24.9 (2.7) | 23.8 (2.6) | 22.2 (2.5) | < .001 | 25.0 (3.3) | 24.4 (3.2) | < .001 |
| Absolute grip strength (kg)[c] | 27.1 (8.7) | 20.8 (6.6) | 25.3 (7.3) | 27.9 (8.0) | 29.5 (8.1) | 32.0 (8.9) | < .001 | 27.3 (8.8) | 27.1 (8.7) | 0.621 |
| Smoking status, n(%) | | | | | | | 0.079 | | | 0.219 |
| Never smoked | 2,261 (80.4) | 462 (82.1) | 461 (82.0) | 454 (80.8) | 448 (79.7) | 436 (77.6) | | 286 (77.1) | 1,975 (80.9) | |
| Former Smoker | 251 (8.9) | 50 (8.9) | 42 (7.5) | 58 (10.3) | 55 (9.8) | 46 (8.2) | | 39 (10.5) | 212 (8.7) | |
| Current smoker | 299 (10.6) | 51 (9.1) | 59 (10.5) | 50 (8.9) | 59 (10.5) | 80 (14.2) | | 46 (12.4) | 253 (10.4) | |
| Current alcohol drinking, n (%)[d] | 1445 (51.4) | 274 (48.7) | 281 (50.0) | 301 (53.6) | 305 (54.3) | 285 (50.7) | 0.274 | 208 (56.1) | 1238 (50.7) | 0.056 |
| Regular exercise, n (%)[e] | 919 (32.7) | 154 (27.4) | 184 (32.7) | 178 (31.7) | 200 (35.6) | 203 (36.1) | **0.013** | 146 (39.4) | 773 (31.7) | **0.003** |
| Living with family, n (%)[f] | 2,491 (88.6) | 486 (86.3) | 504 (89.7) | 485 (86.3) | 501 (89.2) | 515 (91.6) | **0.019** | 336 (90.6) | 2,155 (88.3) | 0.204 |
| ≥High school graduate (%)[g] | 884 (31.5) | 117 (20.8) | 166 (29.5) | 204 (36.3) | 224 (39.9) | 224 (39.9) | < .001 | 94 (25.3) | 790 (32.4) | **0.007** |
| Family history of diabetes, n (%) | 511 (18.2) | 90 (16.0) | 115 (20.5) | 100 (17.8) | 121 (21.5) | 85 (15.1) | **0.020** | 125 (33.7) | 386 (15.8) | < .001 |
| Hypertension, n (%)[h] | 1,621 (57.7) | 371 (65.9) | 332 (59.1) | 325 (57.8) | 312 (55.5) | 281 (50.0) | < .001 | 254 (68.5) | 1,367 (56.0) | < .001 |
| Dyslipidemia, n (%)[i] | 2,175 (77.4) | 462 (82.1) | 460 (81.9) | 451 (80.3) | 436 (77.6) | 366 (65.1) | < .001 | 318 (85.7) | 1,857 (76.1) | < .001 |

Data are presented as mean (SD) unless indicated otherwise.

[a] T2DM was defined as the presence of T2DM (history of physician diagnosis, use of hyperglycemic medication or insulin, or measured fasting glucose ≥126 mg/dL [7.0 mmol/L]).

[b] Weight in kilograms divided by height in meters squared.

[c] Absolute grip strength (kg) was assessed using average value (out of 2 trials of each hand).

[d] Current alcohol drinking was defined as a current drinker and non-current drinker.

[e] Participant regularly participates in exercise enough to initiate sweating.

[f] Participant had lived with family during the last year.

[h] Defined as systolic/diastolic blood pressure ≥ 130/80mmHg, self-reported diagnosed hypertension, and/or taking blood pressure medication.

[i] Defined as if a serum low-density lipoprotein ≥ 130 mg/dL, if a serum total cholesterol ≥ 200 mg/dL, if a serum high-density lipoprotein ≤ 40 mg/dL, or if a serum triglyceride level ≥ 150 mg/dL., self-reported diagnosed dyslipidemia, and/or taking anti-lipidemic medication.

**Table 2. Odds ratios of T2DM by quintiles of relative grip strength.**

| Relative grip strength | N | Cases | OR (95% CI) | | |
|---|---|---|---|---|---|
| | | | Model 1 | Model 2 | Model 3 |
| Q1 (weakest) | 563 | 107 | 1.00 (reference) | 1.00 (reference) | 1.00 (reference) |
| Q2 | 562 | 81 | 0.75 (0.54–1.02) | **0.71 (0.51–0.99)** | 0.73 (0.53–1.02) |
| Q3 | 562 | 70 | **0.64 (0.46–0.89)** | **0.65 (0.46–0.91)** | **0.68 (0.48–0.97)** |
| Q4 | 562 | 72 | **0.68 (0.48–0.95)** | **0.66 (0.46–0.93)** | 0.72 (0.50–1.03) |
| Q5 (strongest) | 562 | 41 | **0.37 (0.25–0.55)** | **0.42 (0.28–0.63)** | **0.48 (0.32–0.74)** |
| P for linear trend | | | < .001 | < .001 | 0.002 |
| Per SD in relative grip strength[a] | | | **0.69 (0.59–0.81)** | **0.72 (0.61–0.84)** | **0.76 (0.64–0.91)** |

[a] A 1 SD in relative grip strength is equivalent to 0.11kg.

Model 1 was adjusted for sex and age (years).

Model 2 was adjusted for Model 1 plus smoking status (never, former, current), current alcohol drinking status (yes or no), regular exercise (yes or no), living with family (yes or no), ≥high school graduate (yes or no), family history of diabetes (yes or no), hypertension (yes or no), dyslipidemia (yes or no).

Model 3 was adjusted for Model 2 plus body mass index (kg/m$^2$).

Compared to the lowest quintile of relative grip strength (weakest), the odds of T2DM were reduced in the upper four relative grip strength quintiles with ORs (95% CIs) of 0.75 (0.54–1.02), 0.64 (0.46–0.89), 0.68 (0.48–0.95) and 0.37 (0.25–0.55), respectively, after controlling for age and sex in model 1 (Table 2). After further adjustment for potential confounders in model 2, the odds of T2DM were similar and quintiles two to five showed significantly lower odds ratios. In Model 3, however, the addition of BMI weakened the association between relative grip strength and T2DM although a significant linear trend was still observed (p = 0.002). Moreover, the ORs for T2DM per SD increase in relative grip strength was 0.76 (0.64–0.91), after adjustment for potential confounders in model 3.

Our sensitivity analyses using sex-specific tertiles and quartiles of relative grip strength showed similar trends as our presented quintiles. When using quintiles of absolute grip strength, the ORs (95% CIs) of T2DM for the second, third, fourth, and fifth absolute grip strength quintiles in the fully adjusted model were 0.79 (0.56–1.12), 0.95 (0.67–1.34), 0.71 (0.48–1.05), and 0.64 (0.42–0.97), respectively.

Fig 1 shows the odds of T2DM across relative grip strength quintiles, stratified by major confounders. Associations between relative grip strength and the odds of T2DM revealed consistent trends across all strata with no significant interaction effects observed (p>0.05); however, results appeared to be stronger among men, those less than 65 years old, those who ever or currently smoke or currently drink alcohol, and those not participating in regular exercise.

Fig 2 demonstrates the combined association between BMI and relative grip strength on the odds of T2DM. Compared to the 'weak and overweight/obese' group, the 'weak and normal weight', 'strong and overweight/obese', and 'strong and normal weight' groups had lower odds of T2DM with ORs (95% CIs) of 0.68 (0.43–1.09), 0.65 (0.46–0.92) and 0.49 (0.35–0.67), respectively. When the joint association analysis was performed using the BMI cut-off points for the Asian population, similar results were observed. Compared to the 'weak and overweight/obese' group, the 'strong and overweight/obese', 'weak and normal weight', and 'strong and normal weight' groups had ORs (95% CIs) of 0.62 (0.46–0.83), 0.80 (0.41–1.56), and 0.58 (0.42–0.81), respectively.

## Discussion

This study examined the association between relative grip strength and the prevalence of T2DM independent of and combined with BMI. The main findings were: 1) higher relative

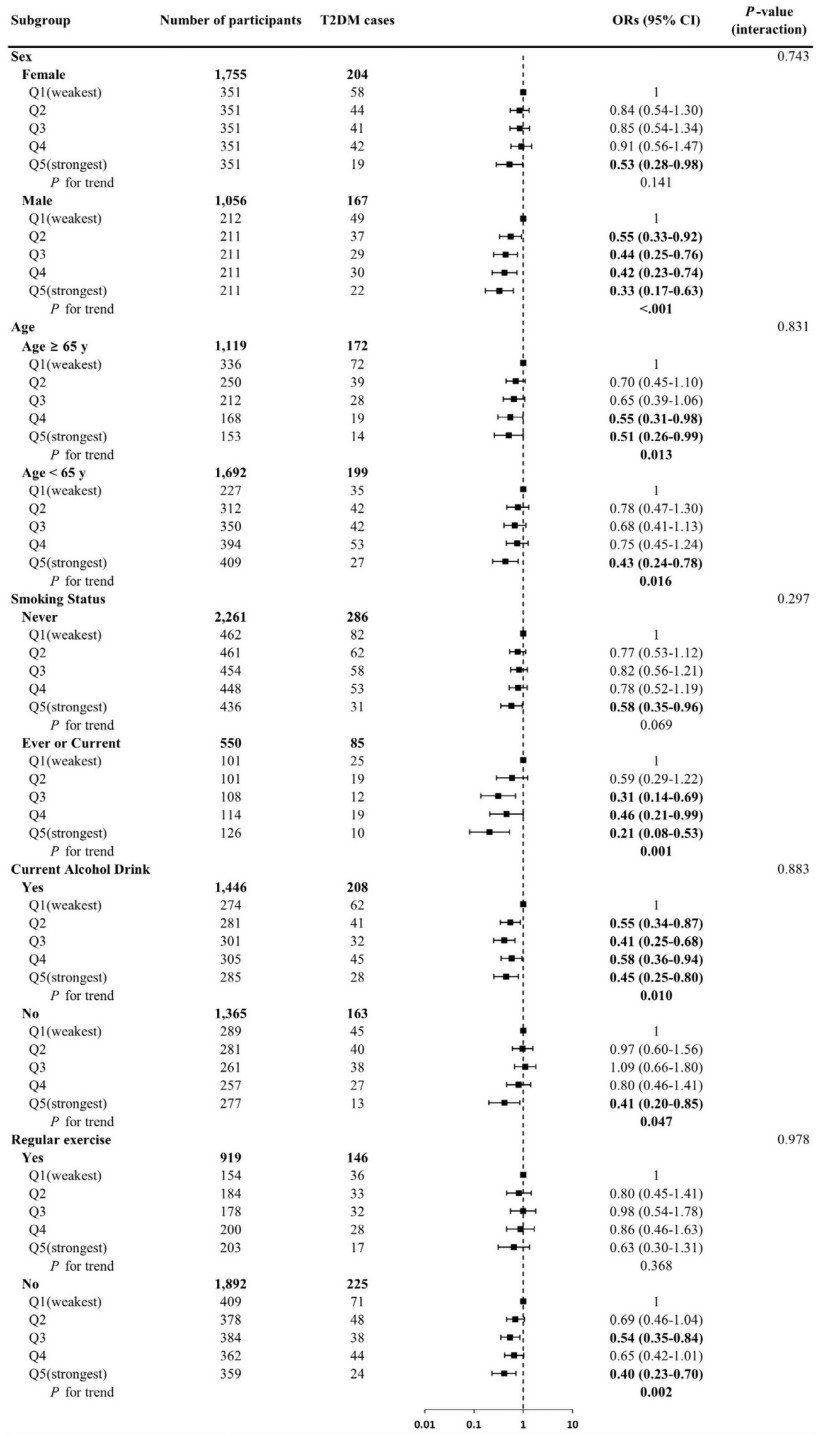

**Fig 1. Odds ratios of T2DM by relative grip strength in stratified subgroup analyses.** Data presented as adjusted odds ratios (95% confidence intervals) across the quintiles of relative grip strength. Analyses were adjusted for sex (not in sex-stratified analyses), age (years, not in age-stratified analyses), smoking status (never, former, or current, not in smoking-stratified analyses), current alcohol drinking (yes or no, not in alcohol-stratified analyses), regular exercise (yes or no, not in regular exercise-stratified analyses), living with family (yes or no), ≥high school graduate (yes or no), family history of diabetes (yes or no), hypertension (yes or no), dyslipidemia (yes or no), and body mass index (kg/m$^2$).

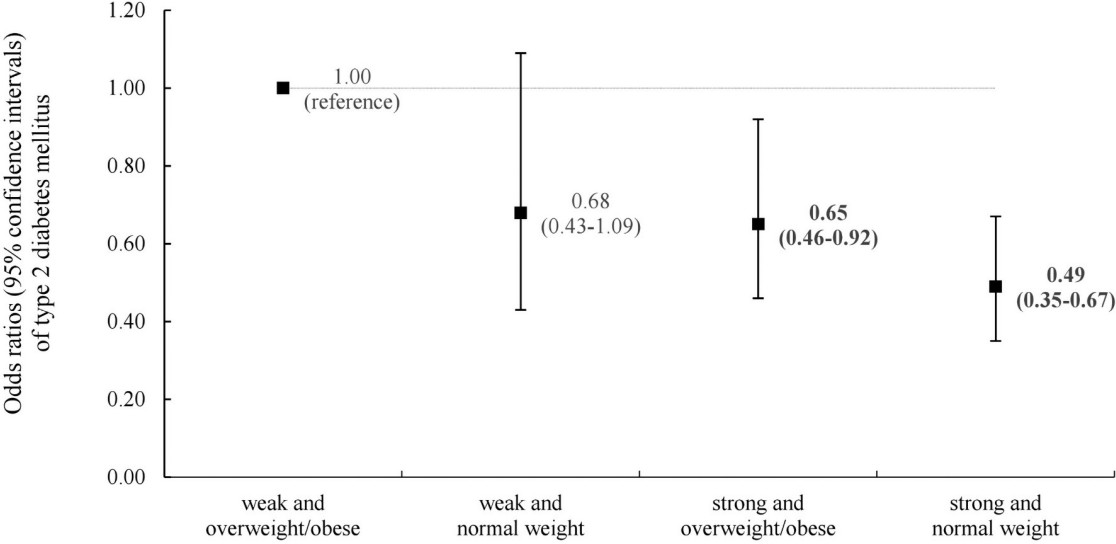

**Fig 2. Joint associations of relative grip strength and body mass index with T2DM.** Participants were divided into four groups based on combined categories of relative grip strength (weak or strong) and body mass index (normal weight, overweight/obese), respectively. "Weak" was the lower 20% of relative grip strength and "strong" was the upper 80% of relative grip strength. Normal weight was BMI $< 25.0 kg/m^2$, overweight/obese was BMI $\geq 25.0 kg/m^2$. The model was adjusted for sex, age (years), smoking status (never, former, current), current alcohol drinking (yes or no), regular exercise (yes or no), living with family (yes or no), $\geq$high school graduate (yes or no), family history of diabetes (yes or no), hypertension (yes or no), dyslipidemia (yes or no), The number of participants (cases of T2DM) in the "overweight/obese and weak," "overweight/obese and strong," "normal weight and weak,", and "normal weight and strong" groups were 352 (73), 824 (120), 211 (34), and 1,424 (144), respectively.

grip strength is associated with the lower prevalence of T2DM independent of BMI (Table 2); 2) the associations between relative grip strength and T2DM were generally consistent across different sex, age groups, and the status of smoking, alcohol drinking, and regular exercise although many subgroups showed no significance partially due to the smaller T2DM cases (Fig 1); and 3) in the joint analysis, stronger relative grip strength was associated with a low prevalence of T2DM regardless of BMI (Fig 2).

The inverse relationship observed between relative grip strength and the prevalence of T2DM in the Korean adult population in the current study is in line with previous studies in other populations [24, 30, 34]. Furthermore, a significant association between relative grip strength and incidence of diabetes has also been shown in prospective studies. In a study of over 21,000 Japanese aged 20 to 92 individuals in the lowest quartile of relative grip strength had 56% greater risk of developing diabetes compared to the highest relative grip strength group [23]. Similarly, following 16-years of follow-up in 440 women, greater relative grip strength was associated with a lower risk of incident diabetes [7]. These results support our findings of greater relative grip strength and lower odds of T2DM.

BMI is commonly used to define obesity [26], with higher BMI associated with increased risk of mortality [35], cardiovascular disease [36], and diabetes [37]. About 90% of T2DM is related to excessive weight [38], so it was expected that the association between relative grip strength with T2DM would be affected by BMI. In several previous studies, higher relative grip strength was associated with a lower prevalence of hypertension [39] and cardiometabolic risk factors [40], but the associations disappeared after controlling for BMI. In our study, however, the association between relative grip strength and T2DM remained even after controlling for BMI. Interestingly, in the prospective studies previously discussed, after adjustment for BMI and waist-to-hip ratio, respectively, relative grip strength remained associated with the

incidence of T2DM [7, 23]. These results in combination with ours suggest there is a significant relationship between muscle strength and T2DM independent of BMI.

In our joint analysis (Fig 2), stronger relative grip strength was associated with a lower prevalence of T2DM regardless of BMI. Conversely, when relative grip strength was weak even with normal weight, the prevalence of T2DM was not significantly attenuated. The results of this study are inconsistent with previous studies that analyzed the joint association between relative grip strength and BMI. A cross-sectional study in Korean adults found that lower absolute grip strength was associated with a higher prevalence of diabetes only in the non-obese population [41]. Furthermore, as a result of a 10-year follow-up of 394 Japanese-American adults, high baseline absolute grip strength significantly lowered diabetes risk but only in leaner individuals [18]. These studies suggest the association between absolute grip strength and diabetes risk is significantly modified by BMI, but this contrasts our findings that higher relative grip strength lowers the prevalence of T2DM in both normal and overweight/obese individuals. One possible explanation of the contradicting findings may be that both prior studies used absolute grip strength without considering the strong association between body weight and grip strength.

In the present study, we relativized grip strength to body weight, as has been done in many previous studies involving muscular strength [8, 42]. As expected, when absolute grip strength was used, the association with T2DM in Model 3 was weaker than when using relative grip strength. This may be due to the increased confounding effect of body weight on both the exposure (absolute grip strength) and the outcome (T2DM) whereas relative grip strength reduces one source of confounding through the exposure. Previous studies have shown that relative grip strength predicted metabolic syndrome better than absolute grip strength [8, 12]. Thus, relative grip strength may be more strongly associated with the prevalence of T2DM prevalence than absolute grip strength.

Relative grip strength may be more closely related to the ratio of muscle mass and fat mass that make up body weight. Muscle strength is highly associated with muscle mass rather than fat mass [43]. Therefore, someone with high relative grip strength adjusted by body weight may have a high proportion of muscle mass that constitutes their body mass and may have less fat mass, whereas someone with low relative grip strength may have a relatively small proportion of muscle mass and a relatively large proportion of fat mass. In our data, high relative grip strength was associated with low body weight and BMI. In other words, the association between high relative grip strength and low body fat may be a factor that explains the association with T2DM. However, there was a difference in the results of analysis when using the relative grip strength normalized by body weight or BMI. In our results, the associations were attenuated when using BMI-normalized grip strength (S2 Table) compared to body weight-normalized grip strength on the prevalence of T2DM. One previous study has reported that weight-normalized grip strength has a greater association with metabolic syndrome than BMI-normalized grip strength [12]. There is no study comparing weight and BMI normalized grip strength on the prevalence of T2DM. Therefore, further studies are clearly warranted to examine the difference between relative grip strength normalized by body weight or BMI.

The mechanistic evidence for the relationship between grip strength and T2DM is still limited, but it can be partially explained by the following evidence. Strength training improves the glycemic control process. Repetitive strength exercise increases the expression of GLUT4, which is a glucose transporter playing an important role in glucose uptake [44]. Additionally, as glucose is used as the main fuel for skeletal muscle, some evidence indicates that it is related to skeletal muscle mass. Muscle strength is highly associated with muscle mass [43], and loss of muscle mass due to physical inactivity or aging causes an inflammatory reaction and increases insulin resistance, leading to diabetes [45]. Conversely, diabetes induces muscle protein

degradation and attenuates the mitochondrial function in muscles, accelerating the loss of muscle mass and strength due to aging. Indeed, a bi-directional Mendelian randomization study to investigate the effects of markers of grip strength on T2DM showed that higher grip strength was associated with lower T2DM risk and conversely showed that T2DM was associated with lower grip strength [46]. As several studies have reported a significant association between grip strength and whole-body strength [47–49], it is possible to partially explain the relationship between grip strength and T2DM based on this evidence. But further studies are needed to explain a clear mechanism.

We performed a subgroup analysis to determine whether the association between relative grip strength and T2DM varies according to each confounder (S3 Table). Although there were no interaction effects in all strata, all subgroups showed some differences in association according to classification, especially when stratified by sex, there was a significant trend only in men. These differences are difficult to explain clearly. One possible explanation is that the different glycaemic profiles between men and women. Impaired fasting glucose more common in men than women, and impaired glucose tolerance are more common in women [50, 51]. So, differences in the association between men and women may be affected by the diagnostic criteria for T2DM used in our study. However, further studies, especially prospective studies, with larger samples are clearly warranted to examine the sex-specific difference relationship between relative grip strength and T2DM.

A strength of our study is that, to our knowledge, it is the first to examine the association between relative grip strength and T2DM, independently of BMI, in a large-scale cohort for the Korean population. This study also utilized comprehensive data analyses including stratified analyses to explore effect modification (Fig 1), joint stratification based on both relative grip strength and BMI (Fig 2), and several important sensitivity analyses. Further, grip strength is an emerging easy, safe, and well validated assessment. However, the study presents some limitations. First, this study is cross-sectional in design, therefore it is not possible to establish a causal relationship between relative grip strength and the prevalence of T2DM, and reverse causality (e.g., T2DM could induce the loss of muscle mass and strength) is possible. Second, our study was limited to Korean adults who lived in Yangpyeong City. Therefore, the results of our study may only be generalizable to adults living in rural areas in Korea or other Asian population, especially Northeast Asians such as Chinese or Japanese people [52]. Third, major covariates that could affect diabetes prevalence, such as socioeconomic status and caloric intake, were not available in this study. Finally, many confounders were collected through a self-report questionnaire thus introducing the potential for information bias and bias due to social desirability, which could subsequently lead to misclassification. This potential bias should be considered when interpreting the results from this study. However, we tried to potentially counter the limitation by setting that the main exposure was measured and that T2DM was determined not only based on self-report of the disease, but also medications and fasting glucose.

In conclusion, relative grip strength was inversely associated with T2DM among Korean adults, independent of BMI, and stronger relative grip strength was associated with a low prevalence of T2DM even with obesity. This suggests that greater muscle strength may be an important factor underlying T2DM regardless of BMI. Prospective studies are needed to examine whether strength training can be an effective method for T2DM prevention and treatment.

## Supporting information

**S1 Table. Odds ratios of T2DM by tertiles of relative grip strength.**
(DOCX)

**S2 Table. Odds ratios of T2DM by quintiles of grip strength divided by body mass index.**
(DOCX)

**S3 Table. Odds ratio of prevalence of type 2 diabetes mellitus, stratified by relative grip strength in subgroup.**
(DOCX)

## Acknowledgments

We thank all the participants and research staffs of the Yangpyeong cohort of Korean Genome and Epidemiology Study for their helpful cooperation in this study.

## Author Contributions

**Conceptualization:** Geon Hui Kim, Bong Kil Song, Yeon Soo Kim.

**Data curation:** Geon Hui Kim, Bong Kil Song, Jung Woon Kim.

**Formal analysis:** Geon Hui Kim, Bong Kil Song, Jung Woon Kim.

**Funding acquisition:** Yu-Mi Kim, Mi Kyung Kim, Bo Youl Choi.

**Investigation:** Geon Hui Kim, Yu-Mi Kim, Mi Kyung Kim, Bo Youl Choi.

**Methodology:** Geon Hui Kim, Bong Kil Song.

**Project administration:** Bong Kil Song.

**Resources:** Yu-Mi Kim, Mi Kyung Kim, Bo Youl Choi.

**Supervision:** Bong Kil Song.

**Validation:** Geon Hui Kim, Bong Kil Song, Duck-chul Lee.

**Visualization:** Geon Hui Kim, Bong Kil Song.

**Writing – original draft:** Geon Hui Kim, Bong Kil Song.

**Writing – review & editing:** Bong Kil Song, Elizabeth C. Lefferts, Angelique G. Brellenthin, Duck-chul Lee, Yu-Mi Kim, Mi Kyung Kim, Bo Youl Choi, Yeon Soo Kim.

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
