## [Decision Letter · Decision Letter 0]

5 May 2021

PONE-D-21-08378

Associations between relative grip strength and type 2 diabetes mellitus: The Yangpyeong cohort of the Korean genome and epidemiology study

PLOS ONE

Dear Dr. Song,

Thank you for submitting your manuscript to PLOS ONE. After careful consideration, we feel that it has merit but does not fully meet PLOS ONE’s publication criteria as it currently stands. Therefore, we invite you to submit a revised version of the manuscript that addresses the points raised during the review process.

It is an interesting study, but as both reviewers pointed out, there are some limitations in the current statistical analysis, the presentation of the paper in the introduction and discussion. Please revise the manuscript accordingly.

We look forward to receiving your revised manuscript.

Kind regards,

Jie V Zhao

Academic Editor

PLOS ONE

Journal Requirements:

In ethics statement in the manuscript and in the online submission form, please provide additional information about the patient records/samples used in your retrospective study. Specifically, please ensure that you have discussed whether all data/samples were fully anonymized before you accessed them and/or whether the IRB or ethics committee waived the requirement for informed consent. If patients provided informed written consent to have data/samples from their medical records used in research, please include this information.

Thank you for stating in your Funding Statement:

This work was partly supported by the Research Program funded by the Korea Centers for Disease Control and Prevention (2007-E71002-00, 2008-E71004-00, 2009-E71006-00, 2010-E71003-00, 2011-E71002-00, 2012-E71007-00, 2013-E71008-00, 2014-E71006-00, 2014-E71006-01, 2016-E71001-00).

We note that you have indicated that data from this study are available upon request. PLOS only allows data to be available upon request if there are legal or ethical restrictions on sharing data publicly. For information on unacceptable data access restrictions, please see http://journals.plos.org/plosone/s/data-availability#loc-unacceptable-data-access-restrictions.

4a) If there are ethical or legal restrictions on sharing a de-identified data set, please explain them in detail (e.g., data contain potentially identifying or sensitive patient information) and who has imposed them (e.g., an ethics committee). Please also provide contact information for a data access committee, ethics committee, or other institutional body to which data requests may be sent.

4b) If there are no restrictions, please upload the minimal anonymized data set necessary to replicate your study findings as either Supporting Information files or to a stable, public repository and provide us with the relevant URLs, DOIs, or accession numbers. Please see http://www.bmj.com/content/340/bmj.c181.long for guidelines on how to de-identify and prepare clinical data for publication. For a list of acceptable repositories, please see http://journals.plos.org/plosone/s/data-availability#loc-recommended-repositories.

Additional Editor Comments:

This is an interesting study, but as the reviewers pointed out, some concerns over the statistical analysis and presentation of this paper exist. Please address these concerns accordingly.

Reviewers' comments:

Reviewer's Responses to Questions

**Comments to the Author**

1. Is the manuscript technically sound, and do the data support the conclusions?

Reviewer #1: Partly

Reviewer #2: Partly

2. Has the statistical analysis been performed appropriately and rigorously? 

Reviewer #1: No

Reviewer #2: N/A

3. Have the authors made all data underlying the findings in their manuscript fully available?

Reviewer #1: No

Reviewer #2: No

4. Is the manuscript presented in an intelligible fashion and written in standard English?

Reviewer #1: No

Reviewer #2: Yes

5. Review Comments to the Author

Reviewer #1: Review of the manuscript (PONE-D-21-08378)

This is a cross sectional study exploring the relation of relative grip strength and type 2 diabetes risk in a Korean population. The study is straightforward although the Introduction and Discussion could be enhanced for better readability. I have some concerns over the statistical analyses and the way how the exposures were being categorized in the analyses. Please see below for my comments.

Major comments

The Introduction should be revised to capture the key points the study is trying to address. I think the issue with previous studies is the lack of consideration of body mass index which is related to grip strength. It would be helpful for authors to explain why considering relative grip strength and BMI altogether with type 2 diabetes is helpful, especially when relative grip strength takes into account the effect of BMI in the calculation.

Have the authors considered using other body composition metrics as the BMI to calculate relative grip strength? If the authors preferred to use body weight, please also provide justifications (PMID: 20191251; 27559733). This would also have implications for all downstream analyses.

It would be good for the authors to repeat the analyses using the actual values of the relative grip strength to assess robustness of findings by different ways of categorization. This will maximize the use of data rather than categorizing them into tertiles or quartiles.

It would be more preferable to assess potential interactions using formal interaction analyses such as the significance of the interaction term. The low number of diabetes case may also make particular subgroup analyses lack sufficient statistical power. The way how grip strength is being dichotomized appears arbitrary (20%) and require further justification.

It is unclear why the authors considered the use of absolute grip strength given the issue with confounding by BMI or other body composition. To me, that would be inferior to the main analyses. The same applies to the BMI cutoff where I think it would be better to use Asian specific cutoffs in the main analyses. The use of different cutoffs probably would not provide much insight since the differences reflect the varying % of fat and lean mass in Asians and Europeans.

The Discussion is a bit long and I think it is not necessary to describe in details the studies the author compared their findings with.

Minor comments

A previous genetic study suggested a causal role of grip strength in risk of type 2 diabetes and can be discussed in the manuscript (PMID: 30798333).

Line 230: The number of participants in reference 23 should be 0.42 million

Line 303-304: Stratified analyses by major confounders: This appears confusing as I think the main purpose is to use stratified analyses to explore effect modification rather than controlling for confounding?

Line 309-310: Would be good to elaborate more on why there are issues with generalisability.

Reviewer #2: Thank you very much for the opportunity to review the study entitled “Associations between relative grip strength and type 2 diabetes mellitus: The Yangpyeong cohort of the Korean genome and epidemiology study”. The manuscript under review assesses whether relative grip strength is associated with type 2 diabetes mellitus (T2DM) prevalence independently of BMI among Korean elderly.

A major strength of this study is acknowledging the use of relative handgrip strength as a safe and convenient assessment for diabetes risk prediction in a representative ethnic-homogenous population.

However, I have some major concerns as follow:

1. In the introduction, the underlying rationale to explore the association of relative grip strength and T2DM independent of and combined with BMI is not clearly addressed by simply stating that “Previous studies investigating the relationship between grip strength and T2DM have controlled for BMI, however, few have investigated relative grip strength and T2DM independent of and combined with BMI.” Previous studies regard BMI as a potential confounder to adjust for because grip strength is known to be correlated with body size (weight, height and BMI) and BMI is a risk factor for diabetes. Can the authors elaborate why examining the association of interest independent of BMI? What is the rationale behind employing a combined association of RELATIVE GRIP STRENGTH and T2DM with BMI?

2. In the logistic regression model 3, the authors controlled for BMI along with other potential confounders but hastily concluded that the association of interest was independent of BMI. These points need clarification along with solid evidence. Other strategies such as automatic variable selection can be used to support or refute whether adjustment for BMI is required (1). In addition, in line 244-246 "however, the association between grip strength and T2DM remained even after controlling for BMI. This suggests that there is a significant relationship between muscle strength and T2DM independent of BMI.", the sentence lacks reasoning. I think the association remain significant after adjustment for a third variable is not enough to say the relationship is independent of the third variable.

3. In line 43, please be more specific in the sentence “may be more strongly associated with various diseases” as of what diseases were RELATIVE GRIP STRENGTH associated with? Are these diseases also correlated to the outcome of interest, which is necessary to address?

4. Lack of justification for the underlying rationale subsequently leads to a few arguable points in the methods and statistical analysis part. In the logistic regression analysis to estimate the odds ratios (OR) of T2DM with the relative grip strength quintiles, important confounders such as socioeconomic position, nutritional intake were not controlled because they were not available. Additionally, is there a need to adjust for “living status” in Model 2?

Also, I wonder if glycaemic status is another important confounder to consider within the association of interest. As suggested in the reference 12, “relative HGS has stronger associations with metabolic syndrome and its component parameters, including high-density lipoprotein cholesterol (HDLC), TG, fasting glucose, and blood pressure”.

Without considering important confounders can lead to confounding bias while accounting for a non-confounder may introduce spurious associations within the association. Both affect the validity of the study that mask the true association and lead to incorrect results and interpretation. However, even after adjusting for potential confounders, residual confounding remains.

5. In Figure 1, why choose 70 as a cut-off age in stratification analysis? The >70 age group has a small sample size, especially among the strongest RELATIVE GRIP STRENGTH quintile, which may lack statistical power to detect the strength of associations. Also, it would be good to report the sex-specific difference in the association between RELATIVE GRIP STRENGTH and T2DM, like the strength of association in female was not significant as in male.

6. In line 195, “Fig 2 demonstrates the additive effects of combining BMI and relative grip strength on the odds of T2DM.” What do you mean by "additive effects"? The analysis in Figure 2 seems to examine the statistical interaction of BMI on RELATIVE GRIP STRENGTH by stratification among subgroups. Is this an appropriate analysis? I doubt about it to treat BMI as an effect modifier as from the Figure 1 results, it is not certain to claim the association between RELATIVE GRIP STRENGTH and T2DM is independent of BMI. I might have missed this, but what do the stratum-specific ORs by BMI look like?

7. Information bias is another issue not addressed in the discussion. This can occur in measuring and classifying covariates, exposure and outcomes and may lead to misclassification. For example, “Do you regularly participate in exercise that elicits sweat” is a qualitative question with a "Yes/No" answer to define “regular exercise”. Furthermore, inaccuracy in classifying self-reported lifestyle variables on smoking, alcohol drinking status, and medical conditions (e.g. hypertension, dyslipidemia, T2DM) can exist. Social desirability bias, as a form of information bias, cannot be ruled out here.

8. In the discussion, the authors acknowledge that it is not possible to establish a causal relationship in this cross-sectional study and stated that T2DM can induce loss of muscle mass and muscle strength in line 299-301. Therefore, reverse causality bias is possible(2).

9. A complete case analysis is based on the assumption that variables were missing completely at random. This may not be valid sometimes if the missing data is dependent on a variable affecting both T2DM risks and missingness of data. Is there a valid assumption in this study to utilize complete case analysis? To validate the assumption, are there sensitivity analyses using different methods such as inverse probability weight to simulate the missing data?

10. For generalizability, the sentence in line 310-311, “the participants were limited to only one ethnic group and living only in one area, so it is difficult to generalize our findings to the population” needs clarification. What does “the population” refer to?

11. Please provide the results of sensitivity analyses that assess the robustness of statistical models in detail if possible, to substantiate the findings.

Minor comments on the manuscript:

1. In line 167, “The addition of BMI in Model 3..”, please be clearer in the description to additionally adjust for BMI based on Model 2 confounders.

2. For the study population, a reference to the cohort profile of the study population would be helpful when discussing generalizability.

3. The subsection "clinical examination" under the "Material and Methods" section describes not only clinical examination. "covariates" would be a better word.

4. Fig2. The 3D graph does not add values to the visual aids but may be quite confusing. A simple 2-by-2 table would be good for results presentation.

5. Causal language was used sometimes within the text. For example, in line 195 when describing Figure 2, “…the additive effects of combining BMI and relative grip strength…” and line 269, the word “effects” would convey causal connotation.

6. Please be consistent when referring to relative grip strength or absolute grip strength throughout the manuscript to avoid confusion. For instance, (line 242-247) “In our study, however, the association between grip strength and T2DM remained even after controlling for BMI.”

References

1. Causal knowledge as a prerequisite for confounding evaluation: an application to birth defects epidemiology. Am J Epidemiol. 2002 155: 176-184

2. Yeung CHC, Au Yeung SL, Fong SSM, Schooling CM. Lean mass, grip strength and risk of type 2 diabetes: a bi-directional Mendelian randomisation study. Diabetologia. 2019;62(5):789-99.

6. PLOS authors have the option to publish the peer review history of their article (what does this mean?). If published, this will include your full peer review and any attached files.

Reviewer #1: No

Reviewer #2: No

---

## [Author Response · Author response to Decision Letter 0]

25 Jun 2021

We thank the editor and reviewers for their helpful edits and comments, which have improved the manuscript, and we are grateful for the opportunity to provide a revised version for consideration. We have carefully considered, discussed, and responded to each comment below, with direct references to the locations (page and line numbers) of changes in the revised manuscript by using track changes. We have submitted our manuscript with the following three file names: ‘Response to Reviewers’. ‘Revised Manuscript with Track Changes’, ‘Manuscript’.

---

## [Decision Letter · Decision Letter 1]

14 Jul 2021

PONE-D-21-08378R1

Associations between relative grip strength and type 2 diabetes mellitus: The Yangpyeong cohort of the Korean genome and epidemiology study

PLOS ONE

Dear Dr. Song,

Thank you for submitting your manuscript to PLOS ONE. After careful consideration, we feel that it has merit but does not fully meet PLOS ONE’s publication criteria as it currently stands. Therefore, we invite you to submit a revised version of the manuscript that addresses the points raised during the review process.

We look forward to receiving your revised manuscript.

Kind regards,

Jie V Zhao

Academic Editor

PLOS ONE

Journal Requirements:

Additional Editor Comments (if provided):

Reviewers' comments:

Reviewer's Responses to Questions

**Comments to the Author**

1. If the authors have adequately addressed your comments raised in a previous round of review and you feel that this manuscript is now acceptable for publication, you may indicate that here to bypass the “Comments to the Author” section, enter your conflict of interest statement in the “Confidential to Editor” section, and submit your "Accept" recommendation.

Reviewer #1: (No Response)

Reviewer #2: All comments have been addressed

2. Is the manuscript technically sound, and do the data support the conclusions?

Reviewer #1: Yes

Reviewer #2: Yes

3. Has the statistical analysis been performed appropriately and rigorously? 

Reviewer #1: Yes

Reviewer #2: Yes

4. Have the authors made all data underlying the findings in their manuscript fully available?

Reviewer #1: Yes

Reviewer #2: Yes

5. Is the manuscript presented in an intelligible fashion and written in standard English?

Reviewer #1: Yes

Reviewer #2: Yes

6. Review Comments to the Author

Reviewer #1: Thank you for addressing my comments. Please see below for some minor points

- I think it would deserve more description to explain possible differences in association when BMI was used to normalize grip strength instead of body weight alone, where the associations were attenuated (in terms of effect size) when BMI was used.

- Regarding generalizability, I think the key thing is whether studies in Koreans can be applied to other populations, such as Chinese.

Reviewer #2: Thank you very much for addressing all comments and questions raised in my previous review. The revised manuscript is much clearer to me now, along with the supplementary material to further expand on sensitivity analyses.

7. PLOS authors have the option to publish the peer review history of their article (what does this mean?). If published, this will include your full peer review and any attached files.

Reviewer #1: No

Reviewer #2: No

---

## [Author Response · Author response to Decision Letter 1]

7 Aug 2021

We thank the editor and reviewers for their helpful edits and comments, which have improved the manuscript, and we are grateful for the opportunity to provide a revised version for consideration. We have carefully considered, discussed, and responded to each comment below, with direct references to the locations (page and line numbers) of changes in the revised manuscript by using track changes. We have submitted our manuscript with the following three file names: ‘Response to Reviewers’. ‘Revised Manuscript with Track Changes’, ‘Manuscript’.

---

## [Editor Report · Decision Letter 2]

10 Aug 2021

Associations between relative grip strength and type 2 diabetes mellitus: The Yangpyeong cohort of the Korean genome and epidemiology study

PONE-D-21-08378R2

Dear Dr. Song,

We’re pleased to inform you that your manuscript has been judged scientifically suitable for publication and will be formally accepted for publication once it meets all outstanding technical requirements.

Kind regards,

Jie V Zhao

Academic Editor

PLOS ONE
---

## [Editor Report · Acceptance letter]

12 Aug 2021

PONE-D-21-08378R2 

Associations between relative grip strength and type 2 diabetes mellitus: The Yangpyeong cohort of the Korean genome and epidemiology study 

Dear Dr. Song:

I'm pleased to inform you that your manuscript has been deemed suitable for publication in PLOS ONE. Congratulations! Your manuscript is now with our production department. 

Kind regards, 

on behalf of

Dr. Jie V Zhao 

Academic Editor

PLOS ONE